**Data Availability Statement:** All relevant data are within the manuscript and its Supporting Information files.

# Analysis of breast cancer survival in a northeastern Brazilian state based on prognostic factors: A retrospective cohort study

**Adriane Dórea Marques**[1,2]*, **Alex Rodrigues Moura**[1,2], **Evânia Curvelo Hora**[1,2], **Érika de Abreu Costa Brito**[1,2], **Leonardo Souto Oliviera**[2], **Ionara Rodrigues Feitosa**[2], **Flavia Fernandes Freitas**[2], **Marcela Sampaio Lima**[1,2], **Íkaro Daniel de Carvalho Barreto**[3], **Marceli Oliveira Santos**[4], **Angela Maria da Silva**[1,2], **Carlos Anselmo Lima**[1,2,5]

1 Health Sciences Graduate Program, Federal University of Sergipe, Aracaju, Sergipe, Brazil, 2 University Hospital-Federal University of Sergipe/EBSERH, Aracaju, Sergipe, Brazil, 3 Biometrics and Applied Statistics Graduate Program, Federal Rural University of Pernambuco, Recife, Pernambuco, Brazil, 4 CONPREV/ Brazilian National Cancer Institute, Rio de Janeiro, Rio de Janeiro, Brazil, 5 Hospital-based Cancer Registry/ HUSE/SES, Aracaju, Sergipe, Brazil

* adrianemarques19@gmail.com

## Abstract

Breast cancer is a major health problem worldwide. Analysis of breast cancer epidemiology in emerging countries enables assessment of prognostic factors, cancer care quality, and the equity of resource distribution. We aimed to estimate the overall (OS) and cancer-specific survival (SS) of breast cancer patients in the northeastern Brazilian state of Sergipe to identify independent prognostic factors. We analyzed a cohort for the factors age at diagnosis, place of residence, time to treatment, staging, and molecular classification, using the Kaplan–Meier method, log-rank test, Pearson's chi-squared test and Cox regression model. The outcome was the vital status at the end of the study. Our analysis showed an OS probability of 0.72 and an SS probability of 0.75. In multivariate analysis, time to treatment within 60 days, stage IV, and triple-negative classification remained independent prognostic factors for both OS [unadjusted hazard ratio (HRp) 1.50 (1.21; 1.86), HRp 16.56 (8.35; 32.85), and HRp 2.73 (1.73; 4.29), respectively] and SS [HRp 1.43 (1.13; 1.81), HRp 20.53 (9.45; 44.56), and HRp 3.14 (1.88; 5.26), respectively]. Better survival was demonstrated for the following patients: those receiving their first treatment after 60 days, with an OS of 52.5 months (51.2; 53.8) and SS of 53.5 months (52.3; 54.7); stage I patients, with an OS of 58.8 months (57.7; 60.0) and SS of 59.2 months (58.1; 60.3); patients without nodal metastasis, with an OS of 54.2 months (53.0; 55.4) and SS of 55.6 months (54.5; 56.7); and patients with luminal A classification, with an OS of 56.8 months (55.0; 58.5) and SS of 57.8 months (56.2; 59.4). This study identified independent prognostic factors and that OS and SS were lower for patients from Sergipe than for patients in high-income areas. Therefore, determining the profiles of breast cancer patients in this population will inform specific cancer care.

**Funding:** We declare that this research was conducted with the partial support of a Research Development Grant from the Fundação de Apoio à Pesquisa e à Inovação Tecnológica do Estado de Sergipe - FAPITEC/SE Protocol: 019.203.00961/2018-2 (https://fapitec.se.gov.br/), awarded to one of the co-authors, CAL, and used solely to sponsor data collection in the cancer registry. Neither CAL nor any other co-author received any salary from the funder. The funder had no role in study design, data collection and analysis, decision to publish, or preparation of the manuscript. The authors did not receive any specific funding for this work. There was no additional external funding received for this study.

**Competing interests:** The authors have no conflicts of interest to declare.

## Introduction

Breast cancer is a major public health problem because it has high incidence rates with consequent high morbidity and mortality. It is the second most common cancer in women after nonmelanoma skin cancer worldwide, with over two million new cases annually [1–3]. Despite early detection resulting in favorable prognosis, breast cancer is still the leading cause of cancer death among women, especially in economically deprived regions [1,4,5].

In some high-income countries, breast cancer incidence and mortality rates have steadily increased, while in others these rates have decreased [6]. However, in low- and middle-income countries, breast cancer incidence and mortality rates have consistently increased [3,7,8]. According to estimates from the Brazilian National Cancer Institute, breast cancer was the most common type of cancer in women (excluding nonmelanoma skin cancer) in 2020, with over 66,000 cases and a mean age-standardized rate (ASR) of 43.74 per 100,000 women. In Northeast Brazil, an economically deprived region, the mean ASR was 43.74 per 100,000. In Sergipe, a northeastern Brazilian state, the mean ASR was 44.27 per 100,000 [3]. The Brazilian Mortality Information System recorded 16,593 deaths from breast cancer in 2019, and the North and Northeast regions displayed the highest rates [9].

Cancer statistics can reveal the effectiveness of public health policies, the equity of resource distribution, and the impact of predictive factors on survival. Therefore, assessing breast cancer survival can help establish criteria for the objective evaluation of patient prognosis and can contribute to the improvement of cancer control strategies [10].

In Brazil, factors such as study design, calendar year, and region and population studied might explain differences in survival [11–13]. Several other factors, such as staging, age at diagnosis, time from diagnosis to treatment, race, histology, and socioeconomic status, may also play a role; however, their contributions are still uncertain [13–18]. It is also noteworthy that breast neoplasms of similar histological subtype can present different outcomes, which may be consistent with the molecular subtype [18].

Based on these assumptions and considering that survival studies in economically deprived regions are scarce, the present study aimed to identify independent prognostic factors for breast cancer survival in the northeastern Brazilian state of Sergipe and to assess how they influenced survival in the study population.

## Materials and methods

We collected data from a retrospective hospital cohort of breast cancer patients treated in the main cancer facility in the state of Sergipe, Brazil from 2014 to 2010. Patients were followed up for at least 60 months.

We used the hospital-based cancer registry (HCR) database to retrieve information from women with invasive breast cancer diagnosed at either the facility or elsewhere. This facility is the largest referral center for the study population; therefore, a considerable number of advanced cases are registered for management. The HCR personnel input demographic, tumor, and treatment variables into the HCR information system, data obtained solely from medical records.

We selected breast cancer cases using the International Classification of Diseases, Oncology, 3rd edition (ICD-O-3), topographical codes C50.0 to C50.9, and morphological codes, 8050/3, 8211/3, 8480/3, 850_/3, 851_/3, 852_/3, 853_/3, and 854_/3.

To define the vital status and to collect information concerning the date and underlying cause of death, the HCR was used to search the Brazilian Mortality Information System of the Ministry of Health. To complement information concerning death, the HCR was used to access the following databases: 1) the National Deceased Registry (CNF Brazil), 2) the Federal

Revenue Service of Brazil, 3) the Brazilian Electoral System, and 4) the National Health Registry.

The Authorization for Outpatient Procedures database was accessed for additional information concerning staging, molecular classification, hormone therapy, chemotherapy, and radiotherapy. The database of the Authorization Hospital Admissions of the Ministry of Health provided information on clinical and surgical admissions.

The variables used were as defined on the standard tumor registry form [19]. For age at diagnosis, we employed age groups according to the hormonal phases (≤45 years, 45 to 54 years, 55 to 64 years, and ≥65 years). We selected other variables, such as place of residence (whether from the capital or countryside/outside), time from diagnosis to treatment, staging [20], histological type, molecular classification (defined as luminal A, luminal B, HER2/neu-enhanced, or triple-negative after immunohistochemical profiling and Fluorescence In Situ Hybridization (FISH) test whenever necessary), and vital status. The time to treatment was set at ≤60 days and >60 days to conform to Law No. 12 732/2012 [21]. Missing data were considered confounding variables that might influence survival estimates.

## Statistical analysis

We used the Kaplan–Meier method to estimate the survival probability of the cohort and then calculated OS and SS for each time interval as the number of women surviving divided by the total number at risk.

To appraise differences among survival distributions, we applied the log-rank test and checked whether any factor would influence the time to event. The Bonferroni method provided compensation for the effect of multiple comparisons, provided correction for the log-rank test results, and assessed differences among several subgroups of variables to control significance levels by adjusting P values.

To evaluate the effect of multiple independent variables and the burden that some prognostic factors may impose upon the outcome, we resorted to Cox's proportional risk model. The method required evaluation of the independent variables by a univariate analysis and then by a multivariate analysis, identifying hazard ratios, adjusted (HR) and unadjusted (HRp), and 95% confidence intervals. To test the proportional hazards assumptions, we employed the method based on scaled Schoenfield residuals.

Pearson's chi-squared model was used to analyze the differences in proportions between the categorized variables to a 5% significance level. The backward selection method selected variables that would fit the tests. We used R Core Team 2020 to perform all the analyses.

## Ethical considerations

The Research Ethics Committee of the Federal University of Sergipe approved this research. We conducted all methods in accordance with relevant guidelines and regulations. As patient databases remained anonymized, obtaining informed consent was not possible. Consequently, as specified in Resolution number 466, December 12, 2012, of the Ministry of Health of Brazil, the ethics committee granted exemption from the necessity for informed consent. In addition, all data remained confidential to be used exclusively for scientific purposes.

## Results

We included 1,278 women with invasive breast cancer in this analysis. Of these, 966 were alive and 312 had died by the end of follow-up. Considering place of residence, 60.7% of the patients lived in the countryside. The median age at diagnosis was 55 years and patients were distributed similarly among the age groups. Invasive ductal carcinomas were the most frequent breast

neoplasm subtype (90.1%). A high number of patients (47.6%) had their first treatment 60 days after diagnosis. Most patients were stage II (32.2%); however, this information was missing from the medical records of 22.9% of cases. Most of the patients did not have lymph node involvement (42.4%) but, again, the number of missing data points was high (32.4%). Most of the cases (30.6%) had their molecular status determined as luminal B. It should be noted that Ki67 was not stained for in 27.5% of cases, preventing determination as either luminal A or B; therefore, these cases were considered luminal X (Table 1).

We estimated the five-year survival probability of the study cohort as cancer-specific survival (SS) of 0.77 (95% CI 0.74; 0.80) and overall survival (OS) of 0.72 (95% CI 0.69; 0.75) (Figs 1 and 2).

Less favorable survival estimates were produced for patients who had their first treatment within 60 days, with an OS of 48 months (95% CI 46.3; 49,6) and SS of 50.1 months (95% CI

**Table 1. Descriptive statistics and mean overall and cancer-specific survival of the cohort during the study.**

| Characteristic | Findings N (%) | MOS (95% CI) | p-value | MSS (95% CI) | p-value |
|---|---|---|---|---|---|
| **No. of patients** | 1,278 | 50.5 (49.5; 51.5) | | 52.0 (51.1; 53.0) | |
| Deceased | 966 (75.6) | | | | |
| Alive | 312 (24.4) | | | | |
| **Residence** | | | | | |
| Capital | 502 (39.3) | 51.4 (49.9; 52.9) | | 52.2 (50.7; 53.6) | |
| Outside the capital | 776 (60.7) | 49.9 (48.6; 51.2) | | 51.9 (50.7; 53.1) | |
| **Age groups** | | | | | |
| <45 | 356 (27.9) | 50.2 (48.3; 52.1) | 0.772 | 51.0 (49.2; 52.9) | 0.364 |
| 45–55 | 352 (27.5) | 50.8 (48.9; 52.6) | | 51.8 (50.0; 53.6) | |
| 56–65 | 290 (22.7) | 51.0 (49.0; 53.0) | | 53.7 (51.9; 55.4) | |
| >65 | 280 (21.9) | 49.9 (47.7; 52.1) | | 51.9 (49.8; 54.0) | |
| Median age (years) | 55 | | | | |
| **Histology types** | | | | | |
| IDC | 1151 (90.1) | 50.7 (49.6; 51.7) | 0.123 | 52.2 (51.2; 53.2) | 0.101 |
| ILC | 54 (4.2) | 48.4 (43.8; 53.1) | | 50.3 (45.8; 54.8) | |
| Special | 73 (5.7) | 49.3 (44.9; 53.7) | | 50.4 (46.2; 54.7) | |
| **T to T (days)** | | | | | |
| ≤60 | 537 (42.0) | 48.0 (46.3; 49.6) | <0.001 | 50.1 (48.5; 51.7) | 0.001 |
| >60 | 608 (47.6) | 52.5 (51.2; 53.8) | | 53.5 (52.3; 54.7) | |
| Missing | 133 (10.4) | 51.2 (48.0; 54.4) | | 52.7 (49.6; 55.8) | |
| **Staging (TNM)** | | | | | |
| I | 196 (15.3) | 58.8 (57.7; 60.0) | <0.001 | 59.2 (58.1; 60.3) | <0.001 |
| II | 412 (32.2) | 55.5 (54.3; 56.8) | | 56.7 (55.5; 57.8) | |
| III | 348 (27.2) | 46.5 (44.5; 48.5) | | 48.5 (46.5; 50.4) | |
| IV | 29 (2.3) | 34.9 (27.1; 42.6) | | 36.8 (29.0; 44.6) | |
| Missing | 293 (22.9) | 44.0 (41.5; 46.6) | | 46.0 (43.5; 48.5) | |
| **Lym node** | | | | | |
| Absent | 542 (42.4) | 50.5 (48.5; 52.4) | <0.001 | 52.0 (50.1; 53.8) | <0.001 |
| Present | 322 (25.2) | 54.2 (53.0; 55.4) | | 55.6 (54.5; 56.7) | |
| Missing | 414 (32.4) | 45.6 (43.5; 47.7) | | 47.3 (45.2; 49.3) | |
| **Mol Clas** | | | | | |
| luminal A | 140 (11.0) | 56.8 (55.0; 58.5) | <0.001 | 57.8 (56.2; 59.4) | <0.001 |
| luminal B | 391 (30.6) | 53.5 (52.1; 55.0) | | 54.6 (53.2; 56.0) | |
| luminal X | 352 (27.5) | 51.9 (50.2; 53.6) | | 53.1 (51.5; 54.7) | |
| HER2 over | 63 (4.9) | 47.3 (42.3; 52.3) | | 52.5 (48.2; 56.7) | |
| Triple-negative | 201 (15.7) | 43.0 (40.0; 46.0) | | 44.6 (41.6; 47.6) | |
| Missing | 131 (10.3) | 43.9 (39.9; 47.9) | | 46.0 (42.1; 49.9) | |

MOS: Mean overall survival; 95% CI: 95% confidence interval; MSS: Mean cancer-specific survival; Missing: Cancer registrar did not identify data in the medical records; T to T: Time to treatment; TNM: TNM Staging System, 7[th] Edition; Lym node: Lymph node involvement; Mol Clas: Molecular classification; luminal X: Estrogen and/or Progesterone receptor positivity, Ki67 not tested.

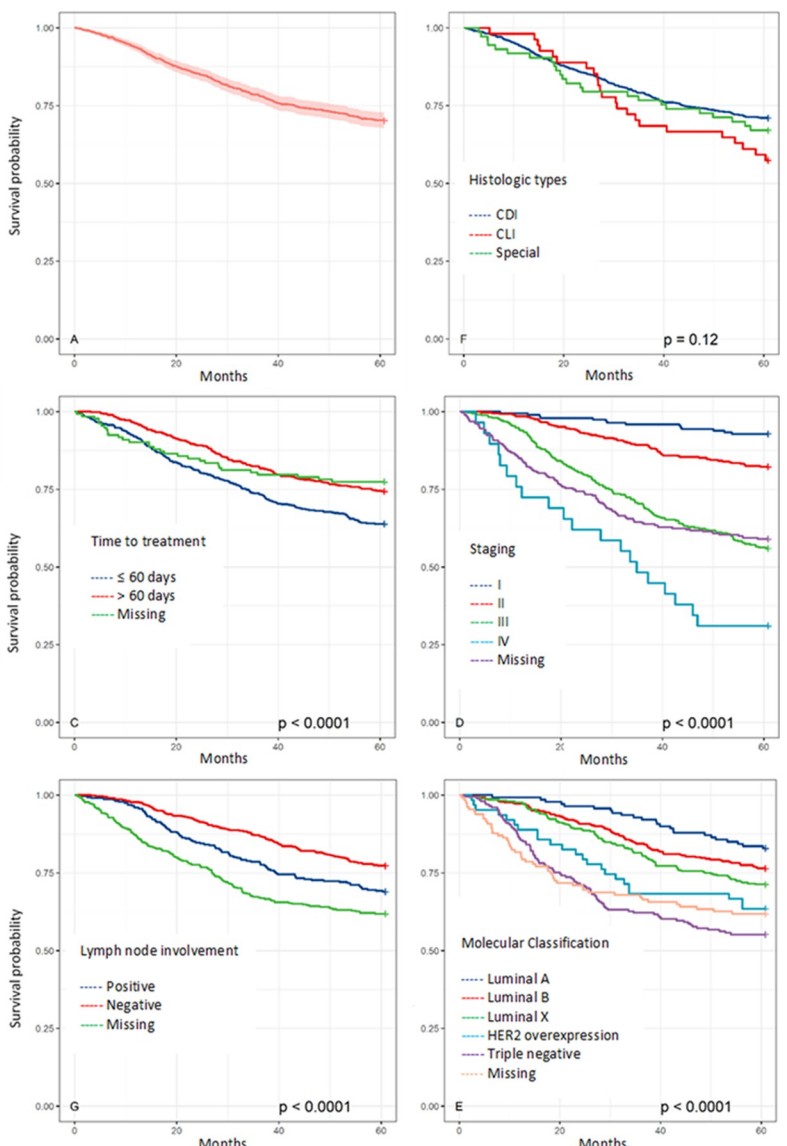

**Fig 1. Overall survival and mean overall survival of breast cancer patients by histology type, time to treatment, staging, lymph node involvement, and molecular classification.**

48.5; 51.7); for patients in stage IV, with an OS of 34.9 months (95% CI 27.1; 42.6) and SS of 36.8 months (95% CI 29.0; 44.6); for patients with lymph node metastasis, with an OS of 50.5 months (95% CI 48.5; 52.4); and for patients with triple-negative classification, with an OS of 43 months (95% CI 40.0; 46.0) and SS of 44.6 months (95% CI 41.6; 47.6). In contrast, better survival estimates were produced for patients who had their first treatment after 60 days, with an OS of 52.5 months (95% CI 51.2; 53.8) and SS of 53.5 months (95% CI 52.3; 54.7); for patients in stage I, with an OS of 58.8 months (95% CI 57.7; 60.0), and SS of 59.2 months (95% CI 58.1; 60.3); for patients without lymph node metastasis, with an OS of 54.2 months (95% CI 53.0; 55.4) and SS of 55.6 months (95% CI 54.5; 56.7); and for patients with luminal A classification, with an OS of 56.8 months (95% CI 55.0; 58.5) and SS of 57.8 months (95% CI 56.2; 59.4) (Table 1).

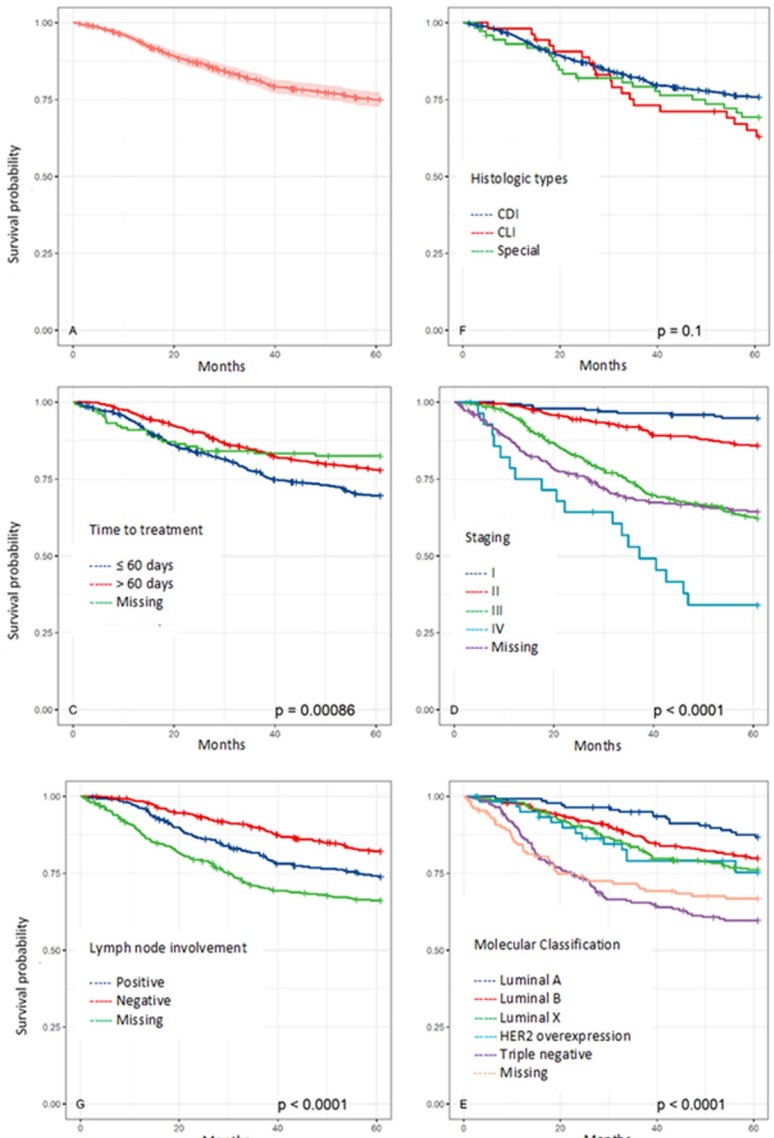

**Fig 2. Cancer-specific survival and mean cancer-specific survival of breast cancer patients by histology type, time to treatment, staging, lymph node involvement, and molecular classification.**

In the univariate (unadjusted) analysis, time to treatment within 60 days, stage IV and triple-negative molecular classification significantly impacted survival. In the multivariate analysis, these variables remained independent prognostic factors for OS [HRp 1.50 (95% CI 1.21; 1.86), HRp 16.56 (95% CI 8.35; 32.85), and HRp 2.73 (95% CI 1.73; 4.29), respectively] and SS [HRp 1.43 (95% CI 1.13; 1.81), HRp 20.53 (95% CI 9.45; 44.56), and HRp 3.14 (95% CI 1.88; 5.26), respectively] (Table 2). Even though the Schoenfield test rejects the hypothesis of hazard proportionality, S1 and S2 Figs show that hazards remain fairly constant throughout the follow-up period, except for the cancer-specific survival variables. Thus, to explain non proportionality, time-dependent variables were presented (Table 3).

**Table 2. Effect of different prognostic factors on the survival probability of patients with breast cancer using the univariate (unadjusted) and multivariate (adjusted) cox regression model.**

| Variable | OVERALL SURVIVAL | | | CANCER-SPECIFIC SURVIVAL | | |
|---|---|---|---|---|---|---|
| | HR (95% CI) | HRp (95% CI)c | p-value | HR (95% CI) | HRp (95% CI) | p-value |
| **Age group** | | | | | | |
| ≤45 | 1.0 | | | 1.0 | | |
| ≤40 | 1.26 (0.98; 1.64) | | | 1.39 (1.05; 1.83) | | |
| ≤55 | 0.97 (0.80; 1.19) | | | 1.16 (0.92; 1.45) | | |
| 46–55 | 0.96 (0.73; 1.26) | | | 0.94 (0.70; 1.25) | | |
| 56–65 | 0.93 (0.70; 1.24) | | | 0.75 (0.54; 1.04) | | |
| >65 | 1.08 (0.82; 1.44) | | | 0.93 (0.68; 1.26) | | |
| **Residence** | | | | | | |
| Outside the capital | 1.0 | | | 1.0 | | |
| Capital | 0.90 (0.73; 1.10) | | | 1.02 (0.81; 1.28) | | |
| **T to T** | | | | | | |
| >60 | 1.0 | 1.50 (1.21; 1.86) | <0.001 | 1.0 | 1.43 (1.13; 1.81) | 0.003 |
| ≤60 | 1.53 (1.24; 1.90) | 0.64 (0.42; 0.97) | 0.036 | 1.47 (1.16; 1.85) | 0.57 (0.35; 0.91) | 0.019 |
| Missing | 0.91 (0.62; 1.34) | | | 0.83 (0.53; 1.29) | | |
| **Staging** | | | | | | |
| I | 1.0 | 2.41 (1.36; 4.28) | 0.003 | 1.0 | 2.62 (1.33; 5.13) | 0.005 |
| II | 2.61 (1.47; 4.63) | 6.97 (4.01; 12.09) | <0.001 | 2.85 (1.46; 5.58) | 7.98 (4.17; 15.27) | <0.001 |
| III | 7.65 (4.42; 13.22) | 16.56 (8.35; 32.85) | <0.001 | 8.75 (4.60; 16.67) | 20.53 (9.45; 44.56) | <0.001 |
| IV | 15.78 (7.97; 31.26) | 6.86 (3.93; 11.97) | <0.001 | 19.66 (9.07; 42.62) | 7.99 (4.16; 15.37) | <0.001 |
| Missing | 7.61 (4.38; 13.25) | | | 8.93 (4.66; 17.10) | | |
| **Lym Node** | | | | | | |
| Missing | 1.0 | | | 1.0 | | |
| Negative | 1.46 (1.12; 1.91) | | | 1.57 (1.17; 2.11) | | |
| Positive | 2.00 (1.57; 2.53) | | | 2.25 (1.73; 2.93) | | |
| **Mol Clas** | | | | | | |
| Luminal A | 1.0 | 1.27 (0.81; 1.99) | 0.307 | 1.0 | 1.0 | 0.200 |
| Luminal B | 1..45 (0.93; 2.28) | 1.25 (0.79; 1.95) | 0.338 | 1.62 (0.97; 2.71) | 1.40 (0.84; 2.34) | 0.280 |
| Luminal X | 1.82 (1.17; 2.84) | 1.95 (1.09; 3.49) | 0.024 | 1.97 (1.18; 3.28) | 1.33 (0.79; 2.22) | 0.253 |
| HER2 over | 2.55 (1.44; 4.52) | 2.73 (1.73; 4.29) | <0.001 | 2.07 (1.03; 4.15) | 1.52 (0.74; 3.11) | <0.001 |
| Triple-Neg | 3.44 (2.19; 5.40) | 2.82 (1.69; 4.71) | <0.001 | 4.00 (2.40; 6.68) | 3.14 (1.88; 5.26) | <0.001 |
| Missing | 2.90 (1.78; 4.71) | | | 3.24 (1.86; 5.63) | 3.18 (1.78; 5.66) | |
| **Histology** | | | | | | |
| IDC | 1.0 | | | 1.0 | | |
| ILC | 1.52 (1.00; 2.32) | | | 1.54 (0.97; 2.45) | | |
| Special | 1.17 (0.77; 1.77) | | | 1.31 (0.85; 2.03) | | |

HR: Unadjusted hazard ratio; HRp: Adjusted hazard ratio; 95%CI: 95% confidence interval; T to T: Time to treatment; Missing: Cancer registrar did not identify data in medical record; Lym Node: Lymph node involvement; Mol Clas: Molecular classification; HER2 over: HER2 overexpression; Triple-Neg: Triple-negative: IDC: Invasive ductal carcinoma; ILC: Invasive lobular carcinoma.

## Discussion

The present study demonstrated that time to treatment, staging, and molecular classification of HER2 significantly impacted OS and SS in the univariate (unadjusted) analysis. In the multivariate (adjusted) analysis, time to treatment after 60 days, stage IV, and triple-negative classification remained independent prognostic factors. The survival estimates observed in the study were lower than those found in some affluent areas of Brazil [22–24], as well as in high-income countries [25,26] and China (89.4%) [27].

Patients in this study under the age of 40 years had a lower OS and SS than older patients. Some studies report that patients under 40 years of age usually present unfavorable prognostic characteristics, usually associated with advanced staging, HER2 overexpression and nodal metastasis [28]. Nixon et al. (1994) reported that women under 35 years of age had poor tumor differentiation, lymphatic involvement, necrosis, and estrogen receptor negativity; consequently, they had

**Table 3. Effect of time-dependent variables on the survival probability of patients with breast cancer using the univariate (unadjusted) and multivariate (adjusted) cox regression model.**

| Variable | HR (95% CI) | HRp (95% CI) | p-valor |
|---|---|---|---|
| **Age group** <br> ≤40 | 1,26 (0,98–1,64) | | |
| ≤55 | 0,97 (0,80–1,19) | | |
| 46–55 | 0,96 (0,73–1,26) | | |
| 56–65 | 0,93 (0,70–1,24) | | |
| >65 | 1,08 (0,82–1,44) | | |
| **Residence** <br> Capital | 0,90 (0,73–1,10) | | |
| Treated | 1,36 (0,94–1,98) | | |
| **T to T** <br> ≤60 | 1,53 (1,24–1,90) | 1,50 (1,21–1,86) | <0,001 |
| Missing | 0,91 (0,62–1,34) | 0,60 (0,40–0,92) | 0,018 |
| **Staging** <br> II | 2,61 (1,47–4,63) | 2,53 (1,43–4,49) | 0,001 |
| III | 7,65 (4,42–13,22) | 7,25 (4,19–12,55) | <0,001 |
| IV | 15,78 (7,97–31,26) | 16,90 (8,52–33,50) | <0,001 |
| Missing | 7,61 (4,38–13,25) | 7,15 (4,10–12,45) | <0,001 |
| **Lym Node** <br> Positive | 1,46 (1,12–1,91) | | |
| Negative | 2,00 (1,57–2,53) | | |
| **Mol Clas** <br> Luminal B | 1,45 (0,93–2,28) | | |
| Luminal X | 1,82 (1,17–2,84) | | |
| HER2 over | 2,55 (1,44–4,52) | | |
| Triplo-Neg | 3,44 (2,19–5,40) | | |
| Missing | 2,90 (1,78–4,71) | | |
| **Histology** <br> ILC | 1,52 (1,00–2,32) | | |
| Non Special | 1,17 (0,77–1,77) | | |

HR: Unadjusted hazard ratio; HRp: Adjusted hazard ratio; 95%CI: 95% confidence interval; T to T: Time to treatment; Missing: Cancer registrar did not identify data in medical record; Lym Node: Lymph node involvement; Mol Clas: Molecular.

more recurrences and distant metastases [29]. In contrast, older women present less aggressive features but have several comorbidities that, when associated with advanced stages, might contribute negatively to survival [13], although this was not shown in our data.

An intriguing finding was that time to treatment ≤ 60 days can be considered as a prognostic factor. While reanalyzing our data, we assumed that this was because of factors such as younger age and more advanced and triple-negative tumors; however, it remained an independent prognostic factor after multivariate analysis. It is possible that improvement in health care plays a role in this finding. The difficulty in accessing diagnosis and treatment in low- and middle-income countries has to be overcome because better cancer survival is a consequence of early diagnosis and timely treatment [30,31].

Advanced staging remained an independent prognostic factor after multivariate analysis, and it became quite clear that both OS and SS decreased as staging progressed. Bulky tumors directly interfere with the quality of life of patients with breast cancer. Conversely, patients presenting at early stages undergo less aggressive modalities of treatment and face a lower risk

of death [32]. In the present study, stage IV indicated an increased risk of death, as also determined by Höfelman et al. (2014) [32] and Fayer (2014) [33].

We were cautious in our analysis because of the high percentage of missing information, mainly in staging, which might have influenced the results. The group with missing data on staging was presented as an independent factor in the multivariate analysis, as also performed by Basílio (2011) [28] and Brito, Portela and Vasconcellos (2009) [34]. Ayala et al. (2019) warned that failure to register this information, especially in the Breast Cancer Information System (SISMAMA), would compromise data monitoring [13].

The different survival probability estimates for different molecular classifications might require different considerations. The different survival probabilities between luminal A and luminal X might be caused by the portion of HER2-overexpressing tumors that were not detected. In addition, missing data (approximately 10%) were shown to be an independent prognostic factor in multivariate analysis, also denoting a confounding factor for survival. Some studies report that a lack of this information may be associated with difficulty in accessing adequate diagnosis and treatment and may be correlated with social status [33,34]. Apart from that, triple-negative classification was a clear independent prognostic factor. Al-Thoubaity (2020) reported that HER2 overexpression and triple-negative features were the most frequently observed; they were associated with an early surge in young women, usually harboring bulky tumors and lymph node metastases [35].

In the present study, the most common histology type was invasive ductal carcinoma, while invasive lobular carcinoma comprised only a small part of the cohort, which was similar to the findings of Basílio (2011) [28]. In our study, lobular carcinomas indicated worse prognosis, which was also observed by others [36–41].

The independent prognostic factors identified in our multivariate analyses were in agreement with hospital-based studies [33,42]. Among these, lymph node metastasis only impacted OS and SS in univariate analysis. Some studies estimated that it indicated an increased risk of death of four to eight times [30,43].

Some limitations of this research should be considered, such as the use of retrospective secondary data from medical records without controls and a lack of standardization of the information in pathological reports. They might have interfered with the accuracy of the presented results.

Despite these limitations, we have determined prognostic factors and estimated the survival probabilities of cancer patients in a northeastern Brazilian state. The outcomes observed indicate the need to improve the cancer care system in this region. The data obtained support the implementation of targeted strategies to improve breast cancer survival irrespective of socioeconomic and cultural background, with the ultimate aim of healthcare equity.

## Conclusions

In summary, our results indicate that independent prognostic factors, such as time to treatment ≤ 60 days, advanced stage, and triple-negative molecular classification, significantly impact OS and SS. In addition, a lack of information, such as staging and molecular classification, may compromise survival analysis and, consequently, jeopardize cancer care actions.

Estimating OS and cancer-specific survival provided a better understanding of the profile of breast cancer patients treated in the state of Sergipe. This emphasizes the need for specific health policies to improve access to cancer facilities for early diagnosis and timely treatment.

## Supporting information

**S1 Fig. The proportional hazards assumptions for the cox model, considering overall survival estimates.**
(TIFF)

**S2 Fig. The proportional hazards assumptions for the cox model, considering specific survival estimates.**
(TIFF)

## Acknowledgments

We thank the following staff of the Hospital Cancer Registry for their excellent work in data collection and database preparation: José Erinaldo Lobo de Oliveira, Elma Santana de Oliveira, Maria das Graças Prata França, Sueli Pina Vieira, Maria Cristina da Conceição Coelho Santos, Marina Ferreira de Oliveira Kobilsek, and Maria das Graças Rodrigues de Melo. We also thank Jeremy Allen, PhD, from Edanz (www.edanz.com/ac) for editing a draft of this manuscript.

## Author Contributions

**Conceptualization:** Adriane Dórea Marques, Carlos Anselmo Lima.

**Data curation:** Íkaro Daniel de Carvalho Barreto.

**Formal analysis:** Adriane Dórea Marques, Íkaro Daniel de Carvalho Barreto, Carlos Anselmo Lima.

**Funding acquisition:** Carlos Anselmo Lima.

**Investigation:** Adriane Dórea Marques, Alex Rodrigues Moura, Evânia Curvelo Hora, Érika de Abreu Costa Brito, Leonardo Souto Oliviera, Ionara Rodrigues Feitosa, Flavia Fernandes Freitas, Marcela Sampaio Lima, Marceli Oliveira Santos, Angela Maria da Silva.

**Methodology:** Adriane Dórea Marques, Alex Rodrigues Moura, Evânia Curvelo Hora, Érika de Abreu Costa Brito, Marcela Sampaio Lima, Carlos Anselmo Lima.

**Software:** Íkaro Daniel de Carvalho Barreto.

**Writing – original draft:** Adriane Dórea Marques, Carlos Anselmo Lima.

**Writing – review & editing:** Adriane Dórea Marques, Carlos Anselmo Lima.

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
