## [Decision Letter · Decision Letter 0]

18 Mar 2021

PONE-D-20-36356

Breast cancer survival analysis based on prognostic fators in a state of Northeastern Brazil: a retrospective cohort study

PLOS ONE

Dear Dr. Marques,

Thank you for submitting your manuscript to PLOS ONE. After careful consideration, we feel that it has merit but does not fully meet PLOS ONE’s publication criteria as it currently stands. Therefore, we invite you to submit a revised version of the manuscript that addresses the points raised during the review process.

Your manuscript has been assessed by two external reviewers, who have raised a number of concerns about the methodology and statistical analyses used. Please ensure that these are addressed in full as part of your revisions.

We look forward to receiving your revised manuscript.

Kind regards,

Joseph Donlan

Senior Editor

PLOS ONE

Journal Requirements:

2. In your ethics statement in the Methods section and in the online submission form, please provide additional information about the data used in your retrospective study. Specifically, please ensure that you have discussed whether all data were fully anonymized before you accessed them and/or whether the IRB or ethics committee waived the requirement for informed consent. If patients provided informed written consent to have data from their medical records used in research, please include this information.

3. Please include the date(s) on which you accessed the databases or records to obtain the retrospective data used in your study.

6. Please upload a copy of Supporting Information Files which you refer to in your text on page 19.

7. Thank you for stating in your Funding Statement:

"CAL - This research was conducted with the partial support of a Research Development Grant from the Fundação de Apoio à Pesquisa e à Inovação Tecnológica do Estado de Sergipe - FAPITEC/SE Protocol: 019.203.00961/2018-2. (https://fapitec.se.gov.br/)"

Reviewers' comments:

Reviewer's Responses to Questions

**Comments to the Author**

1. Is the manuscript technically sound, and do the data support the conclusions?

Reviewer #1: Partly

Reviewer #2: No

2. Has the statistical analysis been performed appropriately and rigorously? 

Reviewer #1: No

Reviewer #2: No

3. Have the authors made all data underlying the findings in their manuscript fully available?

Reviewer #1: Yes

Reviewer #2: Yes

4. Is the manuscript presented in an intelligible fashion and written in standard English?

Reviewer #1: Yes

Reviewer #2: No

5. Review Comments to the Author

Reviewer #1: 1. The fact that the material method is more comprehensive in the abstract, such as the number of patients.

2. Is there contrast between the time between diagnosis and 1st treatment and the result? Shouldn't it be negative to survival as the time between diagnosis and 1st treatment is prolonged? should be discussed with the literature.

3. In the analysis, what is the ratio of unknowns in luminal without ki-67 % information, stage, lymph node, molecular classification. (As far as it is understood, these rates are over 10%.). Isn't it necessary to analyze these unknown data without consideration?

4. Texts in figures are not readable.

Reviewer #2: This would be interesting to see this manuscript published however this is significant and substantial work required before it meets publication standards.

• The Short title does not seem appropriate for the study presented – the aim was to identify prognostic factors. It might be more appropriate to have a Short title, “Breast cancer survival analysis to identify prognostic factors: a retrospective cohort study.

• As this is a single centre study, are referral patterns likely to have influenced these results?

• What is the likely impact of missing data?

• Missing data (for each respective variable) should be reported in the manuscript.

• A secondary aim was to consider the survival according to time to treatment. Surprisingly, women who were treated within 60 days had poorer survival than those treated after 60days.

• Age specific HR reported in Table 3 are confusing. Does the <=55 group also include the <=40 group? What is the reference category here?

o Ideally the reference category for each variable should be included in the Table.

o Are the HR reported (as opposed to the HRp) univariate or multivariate? The results should be clearly explained so there is no doubt for the reader in terms of what they are reading. Presumably the HRp results are the outcome of the backward selection process?

Methods

• Description of the Hospital Registry of Cancer should be included. How are these data collected? Do they include all breast cancer diagnoses and admissions? What variables are collected? Are these medical records, registry data, administrative data?

• Brief and do not seem to describe all methods used.

Statistical Analysis

• No mention of Bonferroni correction used in Table 1

• For the Cox PH models, were the assumptions of this model tested? And how?

Results

• A descriptive table (Table 1) should be provided that provides descriptive statistics of the cohort analysed.

• Kaplan Meier estimates of 5 year survival should be accompanied by confidence intervals to indicate the uncertainty around the estimates.

• Table 1 is very difficult to read and interpret. I would suggest at a minimum that survival days be converted to months or years. Are these median survival estimates?? I would expect to see a table (or two, for each survival outcomes) that reported % 5 yr survival and confidence intervals for each covariate.

• Unclear the purpose of Table 2

• Table titles and figure title should be meaningful and complete, indicating if estimates are adjusted or unadjusted.

• The results of the lymph node status (Table 3) are surprising – I would expect better survival for those who are lymph node negative. Could the authors please confirm these results are correct? Similarly for time to treatment, I would expect women treated earlier to typically have better survival.

Discussion

•

• Ln 242-244: Should this indicate that the study has reported prognostic factors for breast cancer AND overall survival? It is unclear what the reference to time between diagnosis and first treatment refer to – these are not outcomes that have been presented.

• All discussion around the survival outcomes should clearly differentiate between overall survival and breast cancer specific survival, as both these outcomes have been reported.

• Ln253-255: Unclear what results the correlation with time to treatment refers to.?

• LN262-266: Discusses that comorbidity levels and advanced stage contributes to poorer survival by older women however, the results presented in this manuscript indicate that poorer women have poorer breast cancer specific survival, and equivalent overall survival to older women. The authors should discuss and explain this finding.

• Ln270-273: The authors speculate that poor survival in those treated within 60 days may be due to advanced stage and/or triple negative tumours. But if the multivariate (if they are adjusted) results indicate that time to treatment is still a predictor when adjusted for stage and triple negatives, then this does not fully explain why survival is poorer when treated within 60 days.

• LN283 – Does standardise analysis refer to multivariate analysis? The latter would be the preferred term. Standardisation is something very different.

• There is a great focus in the Discussion and Results on the univariate analyses, which are not as important in a study that is looking to identify independent predictors of survival. It would be recommended to focus mainly on the multivariate results in response to the research question.

• Ln322-327: Discussion of missing data is appropriate however, the extent of which has not been quantified in results. It is important to do so.

• Ln 335 indicates that results were stratified, I have seen no evidence of this.

6. PLOS authors have the option to publish the peer review history of their article (what does this mean?). If published, this will include your full peer review and any attached files.

Reviewer #1: No

Reviewer #2: No

---

## [Author Response · Author response to Decision Letter 0]

1 Jun 2021

Response to Reviewers

Reviewer #1: 

1. The fact that the material method is more comprehensive in the abstract, such as the number of patients.

R. We revised and completed material and methods to comply with this observation

2. Is there contrast between the time between diagnosis and 1st treatment and the result? Shouldn't it be negative to survival as the time between diagnosis and 1st treatment is prolonged? should be discussed with the literature.

R. We inserted the following to discuss this point: “An intriguing feature resulting from our estimates was time to treatment ≤ 60 days to be taken as a prognostic factor. After reanalyzing our data, we assumed it was due to factors such as younger age and more advanced and triple-negative tumors; however, it remained an independent prognostic factor after multivariate analysis. Whether improvement in health care would play any role in modifying it can be questioned. The difficulty in accessing diagnosis and treatment in low- and middle-income countries has to be overcome [30] because better cancer survival is a consequence of early diagnosis and timely treatment [31].”

3. In the analysis, what is the ratio of unknowns in luminal without ki-67 % information, stage, lymph node, molecular classification. (As far as it is understood, these rates are over 10%.). Isn't it necessary to analyze these unknown data without consideration?

R. We added this piece in discussion: “We are cautious in analyzing the high percentage of missing information, mainly in staging, which might have influenced the results. The group with missing data on staging was presented as an independent factor in the multivariate analysis, as Basílio (2011) [28] and Brito, Portela and Vasconcellos (2009) [45] confirmed. Ayala et al. (2019) warned that failure to register this information, especially in the Breast Cancer Information System (SISMAMA), would compromise data monitoring [13].”

4. Texts in figures are not readable 

R. We edited figures to a different format.

Reviewer #2: This would be interesting to see this manuscript published however this is significant and substantial work required before it meets publication standards.

R. We had the manuscript edited by Nature Research Editing Service (certificate below)

• The Short title does not seem appropriate for the study presented – the aim was to identify prognostic factors. It might be more appropriate to have a Short title, “Breast cancer survival analysis to identify prognostic factors: a retrospective cohort study.

R. We agreed and changed it.

• As this is a single centre study, are referral patterns likely to have influenced these results?

R. We think so. In fact, we added “We used the hospital-based cancer registry (HCR) database to retrieve information from women with invasive BC diagnosed at either the facility or elsewhere. Since it is the largest referral center for the study population, a considerable number of advanced cases are registered for management” in Materials and Methods.

• What is the likely impact of missing data?

R. We added this piece in discussion: “We are cautious in analyzing the high percentage of missing information, mainly in staging, which might have influenced the results. The group with missing data on staging was presented as an independent factor in the multivariate analysis, as Basílio (2011) [28] and Brito, Portela and Vasconcellos (2009) [45] confirmed. Ayala et al. (2019) warned that failure to register this information, especially in the Breast Cancer Information System (SISMAMA), would compromise data monitoring [13].”

• Missing data (for each respective variable) should be reported in the manuscript.

R. That was reported in results and in Table 1

• A secondary aim was to consider the survival according to time to treatment. Surprisingly, women who were treated within 60 days had poorer survival than those treated after 60days.

R. We inserted the following to discuss this point: “An intriguing feature resulting from our estimates was time to treatment ≤ 60 days to be taken as a prognostic factor. After reanalyzing our data, we assumed it was due to factors such as younger age and more advanced and triple-negative tumors; however, it remained an independent prognostic factor after multivariate analysis. Whether improvement in health care would play any role in modifying it can be questioned. The difficulty in accessing diagnosis and treatment in low- and middle-income countries has to be overcome [30] because better cancer survival is a consequence of early diagnosis and timely treatment [31].”

• Age specific HR reported in Table 3 are confusing. Does the <=55 group also include the <=40 group? What is the reference category here?

o Ideally the reference category for each variable should be included in the Table.

o Are the HR reported (as opposed to the HRp) univariate or multivariate? The results should be clearly explained so there is no doubt for the reader in terms of what they are reading. Presumably the HRp results are the outcome of the backward selection process?

R. We rewrote the results and also the tables to better explained the findings

Methods

• Description of the Hospital Registry of Cancer should be included. How are these data collected? Do they include all breast cancer diagnoses and admissions? What variables are collected? Are these medical records, registry data, administrative data?

R. That was inserted: “We used the hospital-based cancer registry (HCR) database to retrieve information from women with invasive BC diagnosed at either the facility or elsewhere. Since it is the largest referral center for the study population, a considerable number of advanced cases are registered for management. Thus, the HCR personnel input demographic, tumor, and treatment variables into the HCR information system, data which were obtained solely from medical records.”

• Brief and do not seem to describe all methods used.

R. We inserted the following: “We selected BC cases using the International Classification of Diseases, Oncology, 3rd edition (ICD-O-3); topographical codes C50.0 to C50.9; and morphological codes 8050/3, 8211/3, 8480/3, 850_/3, 851_/3, 852_/3, 853_/3, and 854_/3.

To define the vital status and collect information as the date and underlying cause of death, we searched the Brazilian Mortality Information System (SIM) of the Ministry of Health. To complement information on death, we accessed the following databases: 1) National Deceased Registry (CNF Brazil), 2) Federal Revenue Service of Brazil, 3) Brazilian Electoral System, and 4) National Health Registry (NHR).

We also accessed the Authorization for Outpatient Procedures database for additional information concerning staging, molecular classification, hormone therapy, chemotherapy, and radiotherapy. The database of the Authorization Hospital Admissions (AHA) of the Ministry of Health provided us with information on clinical and surgical admissions.”

Statistical Analysis

• No mention of Bonferroni correction used in Table 1

R. We inserted the following: “The Bonferroni method provided compensation for the effect of multiple comparisons, provided correction for the log-rank test results, and assessed differences among several subgroups of variables to control significance levels by adjusting P values.”

• For the Cox PH models, were the assumptions of this model tested? And how?

R. We rewrote the information about the Cox model to try to comply with that: ” To evaluate the effect of multiple independent variables and the burden that some prognostic factor may impose upon the outcome, we resorted to Cox's proportional risk model. First, the method evaluated the independent variables as univariate analysis and then multivariate analysis, identifying hazard ratios, adjusted (HR) and unadjusted (HRp), and 95% confidence intervals.

Pearson's chi-squared model provided analysis of the differences in proportions between the categorized variables to a 5% significance level. The backward selection method selected variables that would fit the tests.”

Results

• A descriptive table (Table 1) should be provided that provides descriptive statistics of the cohort analysed.

R. Table 1 now brings the descriptive statistics of the cohort study

• Kaplan Meier estimates of 5 year survival should be accompanied by confidence intervals to indicate the uncertainty around the estimates.

R. We present confidence intervals.

• Table 1 is very difficult to read and interpret. I would suggest at a minimum that survival days be converted to months or years. Are these median survival estimates?? I would expect to see a table (or two, for each survival outcomes) that reported % 5 yr survival and confidence intervals for each covariate.

R. We redesigned the tables, resulting in different Table 1 and 2. I hope that will meet the concerns presented.

• Unclear the purpose of Table 2

R. Ok, tables were redesigned

• Table titles and figure title should be meaningful and complete, indicating if estimates are adjusted or unadjusted.

R. We redesigned them and hope that will do.

• The results of the lymph node status (Table 3) are surprising – I would expect better survival for those who are lymph node negative. Could the authors please confirm these results are correct? Similarly for time to treatment, I would expect women treated earlier to typically have better survival.

R. Sorry, that was a mistake, as the survival curves for that variable show. That was corrected.

Discussion

•

• Ln 242-244: Should this indicate that the study has reported prognostic factors for breast cancer AND overall survival? It is unclear what the reference to time between diagnosis and first treatment refer to – these are not outcomes that have been presented.

R. That was rewritten to “The present study demonstrated that time to treatment, staging, and molecular classification HER2 significantly impacted OS and SS in the univariate (unadjusted) analysis. In the multivariate (adjusted) analysis, time to treatment after 60 days, stage IV, and triple-negative classification remained independent prognostic factors.”

• All discussion around the survival outcomes should clearly differentiate between overall survival and breast cancer specific survival, as both these outcomes have been reported.

R. Discussion was rewritten and that was done.

• Ln253-255: Unclear what results the correlation with time to treatment refers to.? 

R. We agreed with that and rewrote to “Patients under the age of 40 years had a lower OS and SS in the study. Some studies report that patients under 40 years of age usually present unfavorable prognostic characteristics, usually associated with advanced staging, HER-2 overexpression and nodal metastasis [28]. Nixon et al. (1994) reported that women under 35 years of age had poor tumor differentiation, lymphatic involvement, necrosis, and estrogen receptor negativity; consequently, they had more recurrences and distant metastases [29].”

• LN262-266: Discusses that comorbidity levels and advanced stage contributes to poorer survival by older women however, the results presented in this manuscript indicate that poorer women have poorer breast cancer specific survival, and equivalent overall survival to older women. The authors should discuss and explain this finding.

R. We agreed that was confusing and changed to “In contrast, older women present less aggressive features but have several comorbidities that, when associated with advanced stages, might contribute negatively to survival [13], although this was not shown in our data.”

• Ln270-273: The authors speculate that poor survival in those treated within 60 days may be due to advanced stage and/or triple negative tumours. But if the multivariate (if they are adjusted) results indicate that time to treatment is still a predictor when adjusted for stage and triple negatives, then this does not fully explain why survival is poorer when treated within 60 days.

R. We looked into our date and noticed that the calculation was correct; then, we rewrote to “An intriguing feature resulting from our estimates was time to treatment ≤ 60 days to be taken as a prognostic factor. After reanalyzing our data, we assumed it was due to factors such as younger age and more advanced and triple-negative tumors; however, it remained an independent prognostic factor after multivariate analysis. Whether improvement in health care would play any role in modifying it can be questioned. The difficulty in accessing diagnosis and treatment in low- and middle-income countries has to be overcome [30] because better cancer survival is a consequence of early diagnosis and timely treatment [31].”

• LN283 – Does standardise analysis refer to multivariate analysis? The latter would be the preferred term. Standardisation is something very different.

R. We corrected that.

• There is a great focus in the Discussion and Results on the univariate analyses, which are not as important in a study that is looking to identify independent predictors of survival. It would be recommended to focus mainly on the multivariate results in response to the research question.

R. We rewrote discussion to comply with that.

• Ln322-327: Discussion of missing data is appropriate however, the extent of which has not been quantified in results. It is important to do so.

R. We quantified in results and added the lines in discussion:

 “We are cautious in analyzing the high percentage of missing information, mainly in staging, which might have influenced the results. The group with missing data on staging was presented as an independent factor in the multivariate analysis, as Basílio (2011) [28] and Brito, Portela and Vasconcellos (2009) [45] confirmed. Ayala et al. (2019) warned that failure to register this information, especially in the Breast Cancer Information System (SISMAMA), would compromise data monitoring [13]”; 

“In addition, missing data (approximately 10%) were shown to be an independent prognostic factor in multivariate analysis – in fact, also denoting a confounding factor for survival. Some studies report that a lack of this information may be associated with difficulty in accessing adequate diagnosis and treatment and may be correlated with social status [33,35]”;

 “Some limitations of this research should be considered, such as the use of retrospective secondary data without controls for completeness of information as found in medical records and a lack of standardization of the information in pathological reports. They might have interfered with the accuracy of the presented results.”

• Ln 335 indicates that results were stratified, I have seen no evidence of this.

R. Sorry, that was a mistake. We replaced for “Despite these limitations, it is important to note that the present study estimated the survival probabilities of cancer patients in a northeastern Brazilian state, separated by prognostic factors”.

We thank the reviewers for consideration taken with our manuscript.

---

## [Decision Letter · Decision Letter 1]

1 Jul 2021

PONE-D-20-36356R1

Breast cancer survival analysis based on prognostic factors in a northeastern Brazilian state: A retrospective cohort study

PLOS ONE

Dear Dr. Marques,

Thank you for submitting your manuscript to PLOS ONE. After careful consideration, we feel that it has merit but does not fully meet PLOS ONE’s publication criteria as it currently stands. Therefore, we invite you to submit a revised version of the manuscript that addresses the points raised during the review process.

Please provide additional description and discussion of testing of the proportional hazards assumption for the Cox (proportional hazards) model. Please also proof-read and correct English grammatical errors.

We look forward to receiving your revised manuscript.

Kind regards,

Nancy Lan Guo, Ph.D.

Academic Editor

PLOS ONE

Journal Requirements:

Reviewers' comments:

Reviewer's Responses to Questions

**Comments to the Author**

1. If the authors have adequately addressed your comments raised in a previous round of review and you feel that this manuscript is now acceptable for publication, you may indicate that here to bypass the “Comments to the Author” section, enter your conflict of interest statement in the “Confidential to Editor” section, and submit your "Accept" recommendation.

Reviewer #1: All comments have been addressed

Reviewer #2: (No Response)

2. Is the manuscript technically sound, and do the data support the conclusions?

Reviewer #1: Partly

Reviewer #2: Yes

3. Has the statistical analysis been performed appropriately and rigorously? 

Reviewer #1: Yes

Reviewer #2: No

4. Have the authors made all data underlying the findings in their manuscript fully available?

Reviewer #1: No

Reviewer #2: No

5. Is the manuscript presented in an intelligible fashion and written in standard English?

Reviewer #1: Yes

Reviewer #2: No

6. Review Comments to the Author

Reviewer #1: (No Response)

Reviewer #2: Thank you for addressing my earlier comments. These have been done adequately although the testing of the proportional hazards assumption for the cox (proportional hazards) model needs to be undertaken and/or described.

7. PLOS authors have the option to publish the peer review history of their article (what does this mean?). If published, this will include your full peer review and any attached files.

Reviewer #1: No

Reviewer #2: No

---

## [Author Response · Author response to Decision Letter 1]

2 Sep 2021

We had the final manuscript edited by Jeremy Allen, PhD, from Edanz (www.edanz.com/ac) and added acknowledgment.

Reviewer #1: 

R. We thank you for all the comments addressed previously.

Reviewer #2: Thank you for addressing my earlier comments. These have been done adequately although the testing of the proportional hazards assumptions for the cox (proportional hazards) model needs to be undertaken and/or described.

R. We have described the proportional hazards assumption for the cox model and added S3 and S4 Figs in Supporting information. We also added the comments “To test the proportional hazards assumptions, we employed the method based on scaled Schoenfield residuals”, lines 120-121; and “Even though the Schoenfield test rejects the hypothesis of hazard proportionality, probably because of test sensitivity caused by sample size, Figs S3 and S4 show that hazards remain fairly constant throughout the follow-up period, lines 178-180.

We hope we have answered adequately and thank you for the comments.

---

## [Decision Letter · Decision Letter 2]

14 Sep 2021

PONE-D-20-36356R2Analysis of breast cancer survival in a northeastern Brazilian state based on prognostic factors: A retrospective cohort studyPLOS ONE

Dear Dr. Marques,

Thank you for submitting your manuscript to PLOS ONE. After careful consideration, we feel that it has merit but does not fully meet PLOS ONE’s publication criteria as it currently stands. Therefore, we invite you to submit a revised version of the manuscript that addresses the points raised during the review process.

Please address the reviewer’s comments and provide further clarification on your analysis. Please submit your revised manuscript by November 24, 2021. If you will need more time than this to complete your revisions, please reply to this message or contact the journal office at plosone@plos.org. Please include the following items when submitting your revised manuscript:A rebuttal letter that responds to each point raised by the academic editor and reviewer(s). You should upload this letter as a separate file labeled 'Response to Reviewers'.A marked-up copy of your manuscript that highlights changes made to the original version. You should upload this as a separate file labeled 'Revised Manuscript with Track Changes'.An unmarked version of your revised paper without tracked changes. You should upload this as a separate file labeled 'Manuscript'.If applicable, we recommend that you deposit your laboratory protocols in protocols.io to enhance the reproducibility of your results. Protocols.io assigns your protocol its own identifier (DOI) so that it can be cited independently in the future. For instructions see: https://journals.plos.org/plosone/s/submission-guidelines#loc-laboratory-protocols. Additionally, PLOS ONE offers an option for publishing peer-reviewed Lab Protocol articles, which describe protocols hosted on protocols.io. Read more information on sharing protocols at https://plos.org/protocols?utm_medium=editorial-email&utm_source=authorletters&utm_campaign=protocols.

We look forward to receiving your revised manuscript.

Kind regards,

Nancy Lan Guo, Ph.D.

Academic Editor

PLOS ONE

Journal Requirements:

Reviewers' comments:

Reviewer's Responses to Questions

**Comments to the Author**

1. If the authors have adequately addressed your comments raised in a previous round of review and you feel that this manuscript is now acceptable for publication, you may indicate that here to bypass the “Comments to the Author” section, enter your conflict of interest statement in the “Confidential to Editor” section, and submit your "Accept" recommendation.

Reviewer #2: (No Response)

2. Is the manuscript technically sound, and do the data support the conclusions?

Reviewer #2: Yes

3. Has the statistical analysis been performed appropriately and rigorously? 

Reviewer #2: No

4. Have the authors made all data underlying the findings in their manuscript fully available?

Reviewer #2: No

5. Is the manuscript presented in an intelligible fashion and written in standard English?

Reviewer #2: Yes

6. Review Comments to the Author

Reviewer #2: I am not convinced that the test of slope of schoenfeld residuals is too sensitive given the sample size (which isn't that large). Looking at the graphs of residuals, it appears that the top left graph may reflect non proportionality. Ideally, the authors would include a time dependent variable for that characterisitc. If this interaction term is signficant, that would be provide more evidence of non proportionality, and conveniently, address that at the same time.

7. PLOS authors have the option to publish the peer review history of their article (what does this mean?). If published, this will include your full peer review and any attached files.

Reviewer #2: No

---

## [Author Response · Author response to Decision Letter 2]

24 Nov 2021

Response to Reviewers

We had the final manuscript edited by Jeremy Allen, PhD, from Edanz (www.edanz.com/ac) and added acknowledgment.

Reviewer #1: 

R. We thank you for all the comments addressed previously.

Reviewer #2: I am not convinced that the test of slope of schoenfeld residuals is too sensitive given the sample size (which isn't that large). Looking at the graphs of residuals, it appears that the top left graph may reflect non proportionality. Ideally, the authors would include a time dependent variable for that characterisitc. If this interaction term is signficant, that would be provide more evidence of non proportionality, and conveniently, address that at the same time.

R. After revision with the statistician, we agree with the reviewer’s opinion. So, we are including the lines “Even though the Schoenfield test rejects the hypothesis of hazard proportionality, Figs S3 and S4 show that hazards remain fairly constant throughout the follow-up period, except for the for the cancer-specific survival variables. Thus, to explain non proportionality, time-dependent variables were presented (Table 3)”; and also, are adding Table 3; as can be seen in the revised manuscript with track changes.

---

## [Decision Letter · Decision Letter 3]

17 Jan 2022

Analysis of breast cancer survival in a northeastern Brazilian state based on prognostic factors: A retrospective cohort study

PONE-D-20-36356R3

Dear Dr. Manuscript,

We’re pleased to inform you that your manuscript has been judged scientifically suitable for publication and will be formally accepted for publication once it meets all outstanding technical requirements.

Kind regards,

Nancy Lan Guo, Ph.D.

Academic Editor

PLOS ONE

Additional Editor Comments (optional):

Reviewers' comments:

Reviewer's Responses to Questions

**Comments to the Author**

1. If the authors have adequately addressed your comments raised in a previous round of review and you feel that this manuscript is now acceptable for publication, you may indicate that here to bypass the “Comments to the Author” section, enter your conflict of interest statement in the “Confidential to Editor” section, and submit your "Accept" recommendation.

Reviewer #2: All comments have been addressed

2. Is the manuscript technically sound, and do the data support the conclusions?

Reviewer #2: Yes

3. Has the statistical analysis been performed appropriately and rigorously? 

Reviewer #2: Yes

4. Have the authors made all data underlying the findings in their manuscript fully available?

Reviewer #2: Yes

5. Is the manuscript presented in an intelligible fashion and written in standard English?

Reviewer #2: Yes

6. Review Comments to the Author

Reviewer #2: (No Response)

7. PLOS authors have the option to publish the peer review history of their article (what does this mean?). If published, this will include your full peer review and any attached files.

Reviewer #2: **Yes: **Dr Elizabeth Buckley

---

## [Editor Report · Acceptance letter]

25 Jan 2022

PONE-D-20-36356R3 

Analysis of breast cancer survival in a northeastern Brazilian state based on prognostic factors: A retrospective cohort study 

Dear Dr. Marques:

I'm pleased to inform you that your manuscript has been deemed suitable for publication in PLOS ONE. Congratulations! Your manuscript is now with our production department. 

Kind regards, 

on behalf of

Dr. Nancy Lan Guo 

Academic Editor

PLOS ONE